# Exploring the roles of male partners in the transmission, prevention and control of cervical cancer in Central Kenya: A qualitative study

John H. Mwangi[ID]*, Pretty N. Mbeje, Gloria N. Mtshali

Discipline of Nursing, School of Health Sciences, University of KwaZulu-Natal, Durban, South Africa

* jheowho@gmail.com

## Abstract

### Background

Cervical cancer (CC), primarily caused by persistent infection with high-risk human papillomavirus (HPV) types, remains a major global public health issue. While it primarily affects women, male partners significantly influence HPV transmission, as well as women's access to prevention and treatment through decision-making and social support. However, their role in prevention and treatment engagement remains underexplored.

### Aim

The study aimed to explore the perceptions of couples', healthcare workers', and policymakers' regarding the role of male partners in the transmission, prevention, and control of cervical cancer.

### Setting

The study was carried out in three public county hospitals and community settings in Nyeri, Murang'a and Kirinyaga counties in Central Kenya.

### Methods

We used qualitative research approaches for a comprehensive exploration of the roles of male partners in CC transmission, prevention and control. We conducted in-depth interviews and focus group discussions with 73 participants including 20 couples, 20 Nurses, 2 Clinical officers 2 Gynecologists, 6 Community health workers and 3 County directors of health. All the participants were purposefully sampled. We analyzed data thematically using inductive qualitative analysis approaches.

### Results

Overall, we found key organizing themes including financial and logistical support, moral and emotional support, HPV transmission and prevention, gender norms and

**Data availability statement:** All relevant data are within the manuscript and its Supporting minimal information files.

**Funding:** The author(s) received no specific funding for this work.

**Competing interests:** The authors have declared that no competing interests exist.

beliefs and practices. First male partners were perceived to offer financial support to facilitate cervical cancer treatment services as well as logistical support escorting their partners to the clinic or arranging for transport services. Additionally, men's role in HPV prevention and transmission included supporting their children in vaccination as well as prevention by limiting number of sexual partners. However, Key barriers to male involvement in cervical cancer prevention and treatment reported by participants included limited knowledge about the disease, prevailing cultural beliefs and practices that favor traditional medicine over conventional healthcare, and entrenched gender norms that restrict male participation in reproductive health matters.

## Conclusion

Cervical cancer awareness is limited, and male partners support is shaped by financial, cultural, social, and health system factors. Addressing these issues is key to improving prevention and control efforts.

## Contribution

Identifying the potential barriers and male partners influence in access, screening and treatment of cervical cancer services.

---

## Introduction

Cervical cancer (CC) is the fourth most common cause of cancer incidence and mortality among women globally [1]. CC is the most common cause of cancer and cancer deaths among women in Kenya, and 90% of CC deaths occur in low- and middle- income countries (LMIC) [2]). Cervical cancer accounts for 3,591 (12.2%) of all cancer fatalities and 5,845 (13.1%) of new cancer cases in Kenya each year. It is the second most frequent malignancy in women and the primary cause of cancer-related fatalities in Kenya [3]. Persistent (long-lasting) infection with high-risk human papillomavirus (HPV) types is the necessary cause of virtually all cervical cancers; notably, HPV 16 and 18 together account for approximately 70% of cases worldwide [4]. The 2018 cervical cancer screening guidelines propose to have every woman age 25–49 screened regularly for cervical cancer [5]. Understanding social influences on CC screening and treatment is crucial for better access to medical care.

While cervical cancer primarily affects women, men are not passive onlookers. Studies have shown that men are asymptomatic carriers of HPV, meaning they may not experience any symptoms despite harboring the virus [6]. Additionally, Social factors significantly influence men's role in HPV transmission. Cultural norms, societal expectations, and access to information about HPV and sexual health can shape men's understanding of their role in transmission [7]. Furthermore, limited access to comprehensive sexual education and healthcare services can hinder men's awareness of HPV and its potential consequences [8]. Exploring strategies to augment

knowledge about HPV and cancer screening and fostering trust in the healthcare system among male spouses or partners is essential, particularly with the aim of promoting cervical cancer awareness among immigrants in the USA [9]. Kenya cancer policy 2019–2030 addresses many areas related to cancer prevention and management [10]. However there is paucity of data on partners support and the influence of male partners in the prevention and control of cervical cancer.

Studies have shown that men with multiple sexual partners are at a higher risk of acquiring and transmitting HPV [11,12]. Men also have a substantial influence on women's decisions regarding reproductive health issues including cervical cancer prevention and control [12,13], which could positively impact women's psychological, moral, and financial well-being, but it can also negatively affect them through stigmatization, isolation, or restrictions on healthcare access. Additionally, traditional sexual and reproductive health initiatives have focused primarily on women [14]. However, recognizing the importance of male involvement in these discussions is crucial for promoting acceptance and care uptake [15]. Research suggests that both women and men desire shared decision-making in reproductive matters, and a partner's influence can significantly affect a woman's adherence to reproductive health practices [16]. Spousal support in healthcare matters has a positive influence on health promotion and disease prevention [17]. Exploring the roles and patterns of male partners involvement in cervical cancer transmission, as well as in the uptake of screening and treatment services, can offer valuable insights to enhance program implementation at the facility, county, and national levels in Kenya.

This qualitative study aimed at exploring the roles of male partners in transmission, prevention and control of cervical cancer, as perceived by couples, health care workers and policy makers. This evidence is useful in understanding barriers and facilitators to male partners participation in prevention and treatment of cervical cancer, which could help inform CC prevention and control programs.

## Materials and methods

### Study design

A qualitative research approach with Interpretivism and Constructivism ideologies was adopted to conduct the study [18]. The interpretivism and constructivism paradigm advocate the notion that individuals are intentional and inventive in their actions, actively shaping their social environment. This approach recognizes the dynamic and evolving nature of society, acknowledging the possibility of multiple interpretations of an event influenced by individuals' historical or social perspectives [19].

The relevance of this approach to the study lies in the historical focus of reproductive health programs primarily on women as clients, overlooking the fact that women are not entirely independent of men. Women often have male figures in their lives, such as husbands, fathers, male relatives, and other significant others. Studies, such as the one conducted by Adegboyega et al. [20], have shown that the support of spouses plays a crucial role in influencing the utilization of reproductive health services, particularly those pertaining to cervical cancer prevention and control programs.

A qualitative exploratory, descriptive and contextual research design [21] was employed to explore the roles of male partners as perceived by couples, health care workers and policy makers. The design enabled the participants to share their opinions and experiences regarding male partners involvement in prevention and control of cervical cancer by giving them a platform to express their opinions and experiences.

### Study setting

This study was conducted at three county referral hospitals with Maternal and child health (MCH) clinics and community settings in Central Kenya: Murang'a, Nyeri, and Kirinyaga. The three public health facilities were purposively selected based on previously reported high prevalence and incidence of cervical cancer [22,23]. Additionally, these facilities have recorded more than 2,328 cervical cancer cases between 2017–2020 [24]. All the three facilities offer cervical cancer screening, thermal ablation, cryotherapy treatment and HPV vaccination campaigns [25].

## Study participants and sampling strategy

We conducted in-depth interviews and focus group discussions with 73 participants. These included 20 couples (total of 40 participants) of reproductive age (18–50 years), health care workers who included 20 nurses, 2 clinical officers, two gynecologists, six community health workers and three county directors of health. Purposive sampling was used to select a diverse range of participants who could offer different perspectives and experiences and who were comfortable providing the information needed for the study. Couples were eligible for the interviews if they were of reproductive age and willing to provide consent, health care workers were eligible if they had prior training on cervical cancer screening and treatment and had worked in the clinics for at least six months, community health workers were stationed within local communities, gynecologists and county health directors were deployed in the local counties. The study sample size was guided by information saturation as in the existing literature [26].

Couples were recruited at the MCH clinics after health talks conducted by the study teams. Purposive sampling was used to select couples to participate in the study based on the eligibility criteria and willingness to participate in the study. The recruitment of the various study participants and data collection process took nine weeks in total from 8th April 2024–7th June 2024.

## Data collection

Semi-structured interview guide questions were developed based on a literature review to address the objectives of the study and collect qualitative data. Including, role of male partners involvement in cervical cancer screening and treatment, challenges encountered, and support strategies offered. We conducted in-depth interviews (IDIs) and focus groups discussion (FGDs). The interviews were primarily conducted in Swahili for community health workers, which all participants understood, while English was used for health professionals and directors. Interviews were audio recorded using a digital recorder, transcribed and translated verbatim in English. Interviews were conducted at convenient times for the participants and the research assistants. The health professionals and community health workers were interviewed at their workstations while the county health directors were interviewed in their offices. Data was collected by experienced bachelor-level scientists. The researcher's neutrality provided an open conversation which contributed to rich data collected. In-depth interviews took approximately 30–45 minutes, while focus group discussions took 60–75 minutes. Research assistants wrote field notes after every interview for contextual discussion. Interviews were stopped once information saturation was achieved across the three facilities, and when we observed repetitions and no new themes emerged from the interview.

Three FGDs were conducted involving couples from the selected three counties. The distribution of the couples were as follows; Nyeri and Murang'a seven couples each (28 participants), Kirinyaga six couples (12 participants). The total number of participants were 40 (20 males and 20 females). The date was scheduled 7 days in advance, and all participants were notified two days prior to the interviews. Before the beginning of each FGDs session, the purpose of the study was explained to each couple participants. When all the participants agreed to take part in the discussion, the session was formally started. Considering adequate privacy, silence and adequate lighting, the site for FGDs was selected (one of the MCH rooms as there were no activities going on in the late afternoons). The participants were positioned in an arch shape inside the room. The principal investigator, as a moderator, and research assistant stayed at the front so that everyone would be visible to everybody in a comfortable manner. The participants were encouraged to interact with each other, and some probing questions were used by the PI to make the discussion more effective. Each interview lasted about 60–75 minutes. All IDIs and FGDs were audio recorded with permission, transcribed and translated. However, to supplement the transcripts, the research assistant also took field notes. The discussion was guided by laid down guide and continued until data saturation was achieved. This was evident when responses to the questions began to recur, indicating no emergence of new information. Participants were assured that their identities would be anonymous and information provided would be private and confidential.

## Data analysis

Data were analyzed inductively using the six steps of thematic analysis by Braun and Clarke (2016) [27]. The interview transcripts were first reviewed for completeness, and accuracy in translation among those conducted in Swahili. We then read through all the transcripts for initial theme generation and used rapid ethnographic approaches. We first developed summary templates that included data from both FGDs and IDIs. Secondly, two research team members then completed the summary template using two similar transcripts. The summary templates were then reviewed and discrepancies discussed across the two analysts. After review and discussions, the two team members proceeded to complete memos independently. Third, we then drew matrices in an excel sheet to draw patterns of our emerging themes and wrote down summaries based on the key emerging themes. We checked whether the themes worked in relation to the coded extract and then generated a thematic map. After undertaking separate data analyses, we convened to determine the ultimate themes and sub-themes. Our findings and reporting adhered to the consolidated criteria for reporting qualitative research guidelines (COREQ) [28].

## Trustworthiness of the study

As illustrated in Table 1 the trustworthiness of the study was ensured by establishing confirmability, credibility, transferability, dependability and authenticity[29].

## Ethical considerations

The initial step involved obtaining research approval from the University of KwaZulu-Natal's Biomedical Research Ethics Committee (BREC), reference number BREC/00006580/2023. Additional ethical clearance from the country in which the study is based was secured from Mt. Kenya University's Ethics Review Committee, reference number MKU/ISERC/3433. A license to conduct the research was granted by the National Commission for Science, Technology, and Innovation (NACOSTI) in Kenya, reference number 495117. Authority to collect data was also obtained from the relevant County Government Health offices in the study areas. All participants provided a written informed consent. In all the stages of the study the ethical principles were upheld that included respect for persons, beneficence, integrity, and justice.

## Findings

### Characteristics of the participants

Overall, 20 couples (n = 40) participants participated in the focused group discussion while the rest participated in interviews. The mean age of the couples was 36.9 years for men and 30.5 years for women. The majority (70% n = 28) of the

**Table 1. Trustworthiness strategies for qualitative data.**

| Confirmability | • Member checking and peer debriefing to ensure that data accurately reflected the participant experiences. |
|---|---|
| Credibility | • A thorough collection of data ensured through conducting in-depth interviews.<br>• Member checking employed as a validation method.<br>• Direct quotations from the participants was incorporated.<br>• Field notes were diligently recorded during the interviews. |
| Transferability | • A detailed presentation of the study results was assembled.<br>• Purposive sampling was mainly employed, recognizing the limited scope for generalization in qualitative research. |
| Dependability: | • Peer debriefing sessions was held with proficient and experienced research assistants.<br>• A commitment to consistency was upheld throughout the data collection process.<br>• Audit trail of data analysis procedures and recorded data. |
| Authenticity | • Data was audio recorded<br>• Use of direct quotes from participants to support themes and subthemes. |

couples had secondary education. The mean year for professional practice for nurses, clinical officers and gynecologists was 13 years.

The participants were labelled as $N_n$, $CO_n$, $CHW_n$, $G_n$, $CDH_n$, $FGD1-3_{yn\ or\ xn}$, where N Stands for Nurse, CO for Clinical officer, CHW for Community Health Worker G for Gynecologist, CDH for County director of Health, FGD1–3 for Focus Group Discussion one to three, n for serial number x for female member while y is for male member.

## Themes and subthemes that emerged from the data

Five key themes and eleven subthemes were generated from the data. The findings indicated that the male partners have several roles to play in transmission, prevention and control of cervical cancer. Table 2 Summarizes all themes and sub-themes that emerged from data analysis. Findings are illustrated with quotations stemming from exact participant words from the original text.

## Theme 1: Financial and logistical support

**Subtheme 1.1: Provision of financial support.** Participants reported the significant role of male partners in supporting CC prevention and treatment. Couples reported male partners key roles including providing financial aid with the costs of regular CC screening tests, HPV testing and vaccinations, as well as any necessary treatments, which reduced the financial burden on women. Additionally, some participants emphasized the importance of male partners ensuring that their insurance coverage included provisions for CC screening and treatment. Lack of financial support from male partners often led women to delay or forgo CC screening until sufficient funds were available, which put them at risk of getting into advance stages.

*"Sometimes it is difficult to go to hospital when you don't have money but if you need to do a test for cancer he[partner] gives you money and you feel comfortable to go"* (FGD1$_{x1}$)

**Table 2. Thematic analysis.**

**Objective:** To explore ways in which male partners are involved in cervical cancer prevention and control as perceived by couples, health workers and health planners.

| Question | Theme | Sub theme |
|---|---|---|
| What role do male partners play in their partners' cervical cancer prevention and control procedures? | 1. Financial and logistical support | 1.1. Provision of financial support |
| | | 1.2. Provision of logistical support |
| | 2. Moral support | 2.1. Emotional/psychological support |
| | | 2.2. Positive reinforcement and open communication |
| | | 2.3. Advocacy |
| | 3. HPV Transmission, Prevention and vaccination. | 3.1. HPV transmission and prevention |
| | | 3.2. HPV Vaccination |
| | 4. Cultural norms influencing male partners roles. | 4.1. Gender roles and norms |
| | | 4.2. Beliefs and practices |
| | 5. Health education and health care provision (male as a health care provider) | 5.1. Health education and awareness creation |
| | | 5.2. Health care provision |

Several participants reported varied financial support offered by male partners in cervical cancer screening and treatment. Community health workers highlighted that most men lack knowledge about cervical cancer screening and that hinders their ability to fully support their partners, coupled with merger income challenges. Adult children were also reported as a source of support to their parents in accessing and receiving treatment.

*"Children tend to support their mothers morally and financially more towards seeking health services than their fathers"* (CHW$_2$.)

**Subtheme 1.2: Provision of logistical support.** Participants also reported that male partners offer logistical support to their partners. First, it was reported that some men accompany their wives during clinic visits for screening or treatment services. Among men who do not accompany their partners, it was reported that they ensure that the household chores are well taken care of while their partners attend the clinics to ease their burden. Arranging transport services such as motorbikes and vehicles to take their partners to the hospitals were also mentioned as key areas of support. However, participants reported concerns about laxity in some men supporting their partners. For instance, the burden of childcare and farm work was reported as key barriers to accessing care and treatment.

*"Yes, I agree, it is difficult to manage when you think about leaving all the farm work you are supposed to do and going to the hospital you just decide to stay home otherwise you will not have food, but if he is there and offers to help it becomes easy"* ((FGD2$_{x1}$)

**Theme 2: Moral support**

**Subtheme 2.1: Emotional support.** Participants reported that male partners play a crucial role in supporting cervical cancer prevention and treatment by offering emotional support, understanding, and empathy, which helps reduce stress and anxiety for their partners. Specific ways male partners demonstrated support included accompanying their partners to clinic visits, showing concern for their partners' reproductive health by reminding them of clinic appointments, and adhering to post-treatment care practices, such as abstaining from sexual activity for six weeks following cryotherapy or for a few weeks after thermal ablation. Additionally, female participants echoed the need to have their partners present with them during hospital visits for emotional support.

*"They are our husbands, and they should be there when we are being checked the way they would like us to be there for them in case they are the one being checked"* (FGD2$_{x2}$)

*"He should take me there for support incase am told I have cancer"* (FGD3$_{x3)}$

*"Some of the husbands will come to the clinic to confirm when their wives tell them they have been instructed to abstain from sex for six weeks after cryotherapy"* (N$_2$)

**Subtheme 2.2: Positive reinforcement and open communication.** Respondents also reported open communication as a key support area by male partners. For instance, it was highlighted that men support through encouraging regular screening and adherence to medical advice may enhance a female partners' confidence and motivation. Additionally, maintaining open and honest communication about fears, concerns, and emotions was seen as means to strengthen the couple's bond and improve their ability to cope with the cervical cancer screening and treatment process. Avoiding judgment and stigma, and showing compassion throughout the process, were identified as key aspects of positive reinforcement and support. However, lack of disclosure about the treatment process among some women was seen as a barrier to full access of support that may be offered by their partners.

*"Communication may also be a factor if they are open and have conversations it becomes very easy for them to get support from their spouses but if they keep their diagnosis private you can't help them(N$_8$)"*

**Subtheme 2.3: Advocacy.** Participants emphasized the need for male partners to act as a strong advocate for his partner's healthcare needs, both within the relationship and in interactions with healthcare providers. This can empower the woman and enhance her access to quality services for cervical cancer prevention and control. Sustained male involvement was associated with several facilitating factors, including ensuring that screening services are time-efficient, and their partners received quality care. Men additionally, reported concerns about accompanying their partners to the facilities and lacking a specified place for men, and discomfort sitting with other women attending clinics as a key barrier to the support they may provide.

*"Some of the men who come usually complain of slow-moving queues and nurses going for tea breaks when they haven't been served"* (N$_7$)

*"There was a time I accompanied my wife for our child immunization, and we were clumped together with women who were breastfeeding others very pregnant and the space was small, I felt so uncomfortable'*(FGD2$_{y2}$)

Healthcare providers reported that sustained male partners involvement in cervical cancer prevention and control requires targeted support for both couples and the wider community. Key recommendations included promoting open communication on health issues affecting both men and women, empowering women economically to reduce financial dependence, and increasing government and NGO support for screening initiatives. This support should include funding outreach efforts, ensuring the availability and accessibility of screening services, educating community health workers, and making diagnostic procedures such as Pap smears affordable or free. Additionally, providers emphasized the need for widespread dissemination of accurate information on HPV vaccine safety. A county director of health further noted that the development and implementation of clear policy guidelines are essential for enhancing male partners involvement in cervical cancer control and treatment.

*"With proper guidelines I think ways can be organized to involve the male partners in cervical cancer screening even if it involves integrating it with prostate cancer screening and educating them about cervical cancer"* (CDH$_2$).

## Theme 3: HPV transmission, prevention and vaccination

**Subtheme 3.1: HPV transmission and prevention.** Participants acknowledged that men play a significant role in the transmission and prevention of HPV. Several recognized that men can be asymptomatic carriers, unknowingly transmitting the virus to their female partners through sexual contact, even in the absence of visible symptoms. Participants identified strategies for prevention, including limiting the number of sexual partners and consistent condom use, particularly among those with multiple partners. Notably, few participants had limited knowledge about HPV transmission and why it was a sexually transmitted infection, and were not aware it causes cervical cancer. As one respondent stated, *"Cancer is not an infection how can it be transmitted sexually".* (FGD3$_{y4}$)

*"Many clients coming to our clinics have no basic facts regarding cervical cancer and its prevention"* (N$_8$)

*"Majority of men are not even aware that Cervical cancer can be sexually transmitted"* (N$_{10}$)

*"Majority of male partners are disinterest on their women reproductive health, consider cervical cancer screening as women affair and business"* (CO$_1$)

**Subtheme 3.2: HPV vaccination.** Participants highlighted the important role of male partners, particularly fathers, in supporting HPV vaccination efforts. They acknowledged that the vaccine is good when administered before the onset of sexual activity, emphasizing the need for timely vaccination of children, especially those under the age of 14. Respondents described various ways in which men can promote HPV vaccination, including personally taking their children for vaccination, encouraging other men to do the same, dispelling myths and misconceptions surrounding the vaccine, and supporting their partners in accessing HPV testing. Despite this awareness, many focus group members demonstrated limited knowledge, and widespread myths and misconception about HPV vaccine. For instance, two men were concerned about the efficacy of HPV vaccine and reported that the catholic church stand against vaccination is justified. *"Some people don't have girls, and they are advocating for this vaccine on high note, how stupid are we to forget that the other vaccine for polio 4 years ago? Were it not for the catholic church we could be singing another song now, to hell with their vaccines, mine NO!"*(FGD2$_{y5}$)

Importantly, others were excited to learn about the vaccine during the discussion and wanted to know which facilities offered the vaccination. Participants who had vaccinated their children also used the opportunity to encourage vaccination among other participants. As one respondent highlighted.

*"Mine is vaccinated…two doses, this vaccine is safe…cancer is painful and expensive to treat, kindly let your daughter be vaccinated and she will be safe as mine."*(FGD1$_{x5}$)

### Theme 4: Cultural norms influencing male partners roles

**Subtheme 4.1: Gender norms and roles.** Participants reported that patriarchal cultural norms often limit male involvement in women's health, particularly in reproductive health. In many communities, men are perceived as primary decision-makers, which can restrict women's autonomy in seeking healthcare services. Participants emphasized the need for a shift in these gendered expectations, suggesting that men should lead by example and normalize accompanying their partners to reproductive health clinics. They emphasized that such support should not be viewed as a sign of weakness, but rather as a positive demonstration of shared responsibility and care within relationships.

*"It is difficult for men in this community to accompany their wives here due customs and traditional perceptions"* (N$_{17}$)

**Subtheme 4.2: Beliefs and practices.** Participants identified several cultural and social factors that can hinder effective cervical cancer prevention and control. Reliance on traditional healers was reported as a common practice that often delays timely access to professional medical care. Additionally, stigma and shame associated with sexually transmitted infections, including cervical cancer, were seen as significant barriers preventing men from discussing sexual health openly with their partners or seeking medical advice. Participants also highlighted certain cultural practices like polygamous relationships and wife inheritance as contributing factors to the continued risk of HPV transmission and undermining efforts to prevent and control cervical cancer.

*"Some male partners in this community don't believe in conventional medicine and relies on traditional herbs meaning they can't take their spouses for cervical cancer prevention and control health services"* (CHW$_3$)

### Theme 5: Health education and health care provision

**Subtheme 5.1: Health education and awareness creation.** The importance of men in promoting cervical cancer prevention and control at the family and community levels was also reported. Respondents added that male partners should be sensitized about cervical cancer through the media, peer support, and health campaigns. Highlighting that

this knowledge would equip them better to support their partners. Additionally, a respondent echoed the need for cancer treatment and prevention to be taught in tertiary schools to better support their partners.

"We *can educate the community on the importance of male participation in cervical cancer prevention through Educational campaigns involving the media*" (CO$_2$)

"*In my opinion sex education from junior schools to tertiary educational centers will ensure men are properly informed about cervical cancer and how to mitigate against it*" (N$_6$)

**Subtheme 5.2: Healthcare provision.** Participants highlighted the importance of involving male health care providers in cervical cancer prevention and treatment efforts. They noted that men can contribute by being trained to provide cervical cancer-related health services, conducting screenings, appropriately treating women with cervical lesions, and referring patients to advanced care when necessary. However, policymakers acknowledged existing challenges in ensuring the presence of male health workers, particularly nurses, in cervical cancer screening and treatment services, citing staffing limitations and gender-related barriers within the health system.

"*It is not possible to focus on male trainees as we train those who are already in the clinics and they are mainly women by default*" (CDH$_1$)

"*Our healthcare professionals currently lack adequate skills to effectively handle male clients. I recommend that we incorporate both theoretical and practical training simultaneously to enhance their competency in managing male clients. This approach will better equip them to encourage men to support their wives in reproductive health matters.*" (CDH$_3$)*.*

## Discussion

We used qualitative approaches to explicate male partners roles in cervical cancer treatment and prevention. Our findings highlight the critical role of male partners in cervical cancer treatment and prevention. We found that men play critical roles in supporting women through financial and logistical support. This was highlighted by the need to offer treatment cost as well as to facilitate transport to the health care facilities. Additionally, male partners roles also include moral and emotional support to their partners to go through the treatment or the screening phase. Notably, we also found instances where lack of knowledge or cultural beliefs and practices may hinder the roles that may be played by the male partners highlighting existing societal gaps in cervical cancer treatment and prevention.

We found the crucial financial and logistical support offered by male partners. Similarly, previous research in Uganda highlights that gender power dynamics are patriarchal, with men traditionally controlling family finances and access to healthcare [30]. This support significantly reduces the economic burden on women, and its absence often results in delayed or missed screening, heightening the risk of late-stage diagnosis as seen among low-income women [31]. Financial burden is a key concern that has also been reported among other types of cancers such as breast cancer [32]. Male partners also provided proactive roles by ensuring that insurance coverage are facilitated, as well as offering support with household chores to ensure women can attend to their treatment. Importantly financial support has been shown to improve screening and treatment rates. For instance, a randomized control trial found that women who received financial aid were 20-5 time highly likely to go for CC screening [33]. Beyond financial contributions, logistical support from male partners was described as vital. Especially through accompanying women to clinic visits, handling household chores, and arranging transportation. Given the burden of childcare, women always require support with other studies highlighting key roles played by daughters [34]. Our study reports great potential of male partners providing these roles offered by daughters. Of note is that limited knowledge among partners cause partners to abandon their spouses during treatment

[13]. Given that male partners support improves treatment out comes [35], there is a need to promote male involvement in low-resource settings such as Kenya to enhance treatment outcomes among women.

Men were also reported to play significant roles in HPV transmission and prevention. Participants also acknowledged that men can be asymptomatic carriers, unknowingly transmitting HPV to their female partners, which underscores the importance of male involvement in prevention strategies such as limiting sexual partners and consistent condom use. Similarly, studies have shown that prevention and treatment of HPV among men greatly benefits women [36]. Even without symptoms, men contribute significantly to HPV infection risk among women [37]. However, in our study respondents highlighted key knowledge gaps on HPV transmission and cervical cancer which may be major barriers in cervical cancer prevention and treatment. Knowledge about HPV virus transmission that causes cancer remains limited especially among men. Case in point, a study conducted in Uganda and Kenya showed a mere 22% and 25% knowledge on HPV virus [38,39]. This existing knowledge gap exacerbates male disinterest in women's reproductive care which may hinder prevention efforts [40]. Male partners were also seen as key in decision making by ensuring their children get vaccines prior to sexual debut, accompanying children to appointments as well and encouraging them to adhere to vaccine schedules. Men generally act as primary decision-makers within households regarding HPV vaccination [41]. Importantly, discussions also revealed openness and enthusiasm among some men to learn more and actively participate in vaccination efforts, highlighting an opportunity for targeted education. Innovative strategies to improve HPV vaccination uptake should actively incorporate male involvement in awareness campaigns, with a focus on enhancing prevention support and improving HPV-related health literacy among men.

We also found key traditional and cultural practices that influenced male partners roles. Traditional and cultural practices can potentially influence male partners involvement in cervical cancer treatment and prevention. Similar to hesitancy in supporting women during maternal childcare [42]. These practices can create barriers to understanding, participation, and support for women's health issues including reproductive health. The study identified male masculinity as a barrier in male partners support towards cervical cancer prevention in that some male partners associated accompanying their women to MCH clinics as a weakness. Some men were also reported to have low opinions on conventional health services and were relying more on traditional herbs and other remedies. The male masculinity where accompanying women to the hospital is perceived as a weakness should be discredited. This can be through campaigns and mobilizing the gatekeepers in the community to advocate for male support during cervical cancer prevention and control strategies. The findings are supported by a study in Uganda where cervical cancer screening among married women was significantly associated with intimate-partner factors; women's education attainment, intimate-partner's emotional and financial support. The intimate-partner's factors associated with cervical cancer screening point to traditional marital roles of male partners dominance. Efforts to encourage men's participation through community education was recommended [43,44]. Male partners also play crucial roles in educating family and community in cervical cancer. This is consistent with findings from rural Ghana on male support for cervical cancer screening and treatment [45]. Importantly, our study found that healthcare workers supporting women in cervical cancer screening and treatment are predominantly female. Additionally, participants recommended knowledge among male health care workers to go beyond clinical settings and promote cervical cancer awareness and care at the community level.

## Implications for future research, practice, and policy

Male partners roles described in our study have significant implications in research practice and policy aimed at strengthening cervical cancer prevention and treatment. First, future research may need to explore male partners roles among women accessing cervical cancer screening and treatment to understand actual involvement. Within health care facilities in practice, health care providers should be trained to actively involve male partners in education and care processes. Health education should also address existing myths and cultural beliefs to further strengthen male involvement in cervical cancer care. Within the policy level, male partners engagement is crucial and should be incorporated in national cervical cancer campaigns and mobilization to promote shared responsibility in reproductive health.

## Limitations of the study

This qualitative study provides important findings on male involvement in cervical cancer screening and treatment. However, it is not without limitations. A key limitation is that understanding male partners roles was hypothetical and perceptions may differ with real life experiences with women accessing cervical cancer screening or treatment services. Additionally, participant recruitment was biased among those seeking health care services. As a result, findings may differ among couples in other settings, particularly other resource-limited contexts where health-seeking behaviors and social dynamics are different.

## Conclusion

In conclusion, this qualitative study highlights diverse critical roles male partners play in cervical cancer prevention and treatment as perceived by couples, health care workers and policy makers, encompassing financial, logistical, emotional, and moral support that significantly alleviate the burden on women and improve treatment adherence. However, knowledge gaps and traditional cultural beliefs create significant barriers to male involvement. Key knowledge gaps should be addressed through targeted education and community engagement is essential to strengthen male partners involvement and enhance the effectiveness of cervical cancer prevention and care. Interventions aimed at increasing male partners engagement, fostering supportive communication, and reducing stigma are likely to have a significant impact on improving cervical cancer care uptake and adherence.

## Supporting information

**S1 Table. Demographic characteristics of the study participants and study sites.**
(PDF)

**S1 File. Inclusivity in global research questionnaire.**
(DOCX)

**S2 File. Sample translated focused group discussion transcript.**
(PDF)

**S3 File. Sample line by line coding of a transcript.**
(PDF)

## Acknowledgments

The author would like to acknowledge Dr. Pretty N. Mbeje and Prof. Gloria N. Mtshali for their full support as supervisors in writing this manuscript. Acknowledgement is extended to all the study participants for their cooperation and full engagement.

## Author contributions

**Conceptualization:** John H. Mwangi.

**Data curation:** John H. Mwangi.

**Formal analysis:** John H. Mwangi, Gloria N. Mtshali.

**Funding acquisition:** Pretty N. Mbeje, Gloria N. Mtshali.

**Investigation:** John H. Mwangi.

**Methodology:** John H. Mwangi.

**Project administration:** Pretty N. Mbeje, Gloria N. Mtshali.

**Resources:** Pretty N. Mbeje, Gloria N. Mtshali.

**Software:** Pretty N. Mbeje, Gloria N. Mtshali.

**Supervision:** Pretty N. Mbeje, Gloria N. Mtshali.

**Validation:** Pretty N. Mbeje, Gloria N. Mtshali.

**Visualization:** Pretty N. Mbeje, Gloria N. Mtshali.

**Writing – original draft:** John H. Mwangi.

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
