## [Decision Letter · Decision Letter 0]

13 Jun 2025

Dear Dr. MWANGI,

Thank you for submitting your manuscript to PLOS ONE. After careful consideration, we feel that it has merit but does not fully meet PLOS ONE’s publication criteria as it currently stands. Therefore, we invite you to submit a revised version of the manuscript that addresses the points raised during the review process.

We look forward to receiving your revised manuscript.

Kind regards,

Morufu Olalekan Raimi, Ph.D

Academic Editor

PLOS ONE

**Journal Requirements:**

1. When submitting your revision, we need you to address these additional requirements. Please ensure that your manuscript meets PLOS ONE's style requirements, including those for file naming. The PLOS ONE style templates can be found at https://journals.plos.org/plosone/s/file?id=wjVg/PLOSOne_formatting_sample_main_body.pdf and https://journals.plos.org/plosone/s/file?id=ba62/PLOSOne_formatting_sample_title_authors_affiliations.pdf 2. Please include a complete copy of PLOS’ questionnaire on inclusivity in global research in your revised manuscript. Our policy for research in this area aims to improve transparency in the reporting of research performed outside of researchers’ own country or community. The policy applies to researchers who have travelled to a different country to conduct research, research with Indigenous populations or their lands, and research on cultural artefacts. The questionnaire can also be requested at the journal’s discretion for any other submissions, even if these conditions are not met. Please find more information on the policy and a link to download a blank copy of the questionnaire here: https://journals.plos.org/plosone/s/best-practices-in-research-reporting. Please upload a completed version of your questionnaire as Supporting Information when you resubmit your manuscript. 3. We note that your Data Availability Statement is currently as follows: All relevant data are within the manuscript and its Supporting Information files. Please confirm at this time whether or not your submission contains all raw data required to replicate the results of your study. Authors must share the “minimal data set” for their submission. PLOS defines the minimal data set to consist of the data required to replicate all study findings reported in the article, as well as related metadata and methods (https://journals.plos.org/plosone/s/data-availability#loc-minimal-data-set-definition). For example, authors should submit the following data: - The values behind the means, standard deviations and other measures reported;- The values used to build graphs;- The points extracted from images for analysis. Authors do not need to submit their entire data set if only a portion of the data was used in the reported study. If your submission does not contain these data, please either upload them as Supporting Information files or deposit them to a stable, public repository and provide us with the relevant URLs, DOIs, or accession numbers. For a list of recommended repositories, please see https://journals.plos.org/plosone/s/recommended-repositories. If there are ethical or legal restrictions on sharing a de-identified data set, please explain them in detail (e.g., data contain potentially sensitive information, data are owned by a third-party organization, etc.) and who has imposed them (e.g., an ethics committee). Please also provide contact information for a data access committee, ethics committee, or other institutional body to which data requests may be sent. If data are owned by a third party, please indicate how others may request data access.

**Additional Editor Comments:**

Editor Decision

Decision: Major Revision Required

After careful consideration of the reviewers’ comments, the editorial assessment is that the manuscript addresses an important and underexplored topic, the roles of male partners in the transmission and prevention of cervical cancer in Central Kenya. Both reviewers acknowledge the manuscript’s potential contribution to public health literature in low- and middle-income countries. However, substantial revisions are necessary to address concerns regarding clarity, methodological rigor, consistency, and alignment between objectives, findings, and conclusions.

Editor Comments

1. Title and Abstract

• Revise the title for grammatical accuracy, clarity, and consistency (e.g., use “male partners” and “A Qualitative Study” as suggested)

• The abstract requires grammatical corrections, more specificity regarding male partners’ roles, and a clearer connection to barriers in male involvement and implementation challenges

• Ensure consistency in terminology (e.g., “transmission,” “prevention,” and “control”) throughout the manuscript and between the title, abstract, and aims

2. Background

• Reorganize the background to provide a logical flow: begin with the global and regional burden of cervical cancer, then contextualize the situation in Kenya and LMICs

• Avoid redundancy and ensure all references are current, relevant, and properly cited

• Clearly state the research question or objectives at the end of the background section

3. Methods

• The methodology section requires a complete overhaul for clarity and transparency:

o Clearly describe the qualitative research design, sampling approach, participant selection, and data collection methods (e.g., interviews, focus groups)

o Clarify inconsistencies in participant numbers and sampling procedures, especially regarding couples and the use of both purposive and systematic sampling

o Provide detailed information about the study settings, including characteristics of the hospitals and rationale for their selection

o Explain how data saturation was determined for each participant group

o Ensure all acronyms are defined upon first use

4. Results/Findings

• Rename this section as “Findings” to align with qualitative research conventions

• Reanalyze the data using a recognized qualitative analysis framework (e.g., Braun and Clarke’s thematic analysis) and ensure that themes and subthemes are logically developed and supported by appropriate participant quotes

• Avoid redundancy and overlap between themes; ensure that each theme directly addresses the study objectives

• Clarify the distinction between perceived roles and challenges, and adjust themes accordingly

5. Discussion

• Revise the discussion to:

• Summarize findings in relation to the study objectives

• Compare and contrast findings with existing literature, providing justification for similarities and differences

• Discuss the implications of findings for clinical practice, policy, and future research

• Address methodological limitations more comprehensively

6. Conclusion

• Revise the conclusion to ensure it stems directly from the findings, highlighting the implications for practice and policy rather than merely summarizing the study

• Ensure consistency in terminology and alignment with the revised findings

7. References

• Review and correct all references for accuracy, relevance, and adherence to journal formatting guidelines

• Use a reference manager to ensure proper citation and scientific integrity

8. Language and Formatting

• Thoroughly proofread the manuscript to correct typographical, grammatical, and formatting errors

• Ensure consistent use of terms and proper structure throughout the manuscript

Summary:

The manuscript requires major revisions to address methodological, structural, and editorial concerns. The authors are encouraged to systematically address each reviewer comment, provide a detailed response letter, and submit a thoroughly revised manuscript. Upon satisfactory revision, the study has the potential to make a significant contribution to the literature on cervical cancer prevention in LMICs.

Reviewers' comments:

Reviewer's Responses to Questions

**Comments to the Author**

1. Is the manuscript technically sound, and do the data support the conclusions?

Reviewer #1: No

Reviewer #2: Partly

2. Has the statistical analysis been performed appropriately and rigorously?

Reviewer #1: N/A

Reviewer #2: I Don't Know

3. Have the authors made all data underlying the findings in their manuscript fully available?

Reviewer #1: No

Reviewer #2: Yes

4. Is the manuscript presented in an intelligible fashion and written in standard English?

Reviewer #1: No

Reviewer #2: Yes

**Reviewer #1:**  Thank you for the opportunity to review this manuscript. This is a valuable manuscript with meaningful insights into cancer prevention and management in LMICs. However, some sections require clarification, statistical rigor, and better alignment between objectives, results, and conclusions. Improvements in structure, clarity, and depth of interpretation would enhance its contribution to public health literature. Below are specific comments on the manuscript;

Title

-Consider improving grammatical agreement and clarity of focus to enhance precision and readability.

-Consider changing "male partner" to "male partners" to match plural usage unless the study focused on a single partner per participant.

- Replace “A Qualitative Approach” with “A Qualitative Study” for conciseness.

-You can write it this way or otherwise: "Exploring the Roles of Male Partners in the Transmission and Prevention of Cervical Cancer in Central Kenya: A Qualitative Study"

Abstract

Background

-The phrase “While the it primarily affects women” contains a grammatical error (“the it” is redundant). It should read “While it primarily affects women.”

-The statement “men play a crucial role in the transmission of HPV and also influence the processes of prevention and control” is broad and lacks specificity about how men influence these processes (e.g., behaviors, decision-making).

-The background does not connect to implementation challenges (e.g., barriers to male involvement in prevention programs), which would strengthen its relevance to implementation science.

Aim

-The phrase “The study aimed at exploring couples, health care workers and policy makers perceptions” is grammatically incorrect and lacks possessive form or an apostrophe. It should read “couples’, healthcare workers’, and policymakers’ perceptions.”

-The term “male partner” in the aim is singular, inconsistent with the title’s plural “MALE PARTNER” (noting the title’s error). It should be “male partners” for consistency

Setting

-The setting is hospital-based, but the study includes community health workers and couples, suggesting data collection may extend beyond hospitals (e.g., community settings). This discrepancy needs clarification.

-Specific counties (e.g., Kiambu, Nyeri) are not named, which would enhance precision, especially for a regional study

Methods

-The methods section lacks many key issues like sampling approach, trustworthiness, etc.

-There is no qualitative research design. Please read and reflect on what you did and report accordingly the research approach and design

-The participant breakdown is detailed but inconsistent in formatting (e.g., “20 couples,20 Nurses two clinical officers two gynecologists, six Community health workers and three county directors of health”). It lacks punctuation and capitalization consistency (e.g., “Nurses” vs. “nurses,” “Community health workers” vs. “health workers”).

-The total of 73 participants is stated, but it’s unclear how “20 couples” translates to individuals (e.g., 40 individuals if both partners were interviewed). This needs clarification.

-The methods lack mention of data collection methods (e.g., interviews, focus groups), which is critical for understanding the qualitative approach.

Results

-Please summarize and report only the key findings

-The phrase “Based on the participants perceptions” is grammatically incorrect (missing possessive form). It should read “Based on participants’ perceptions.”

-The results are broad and lack specific examples or depth (e.g., what myths/misconceptions? what cultural practices?). This limits the abstract’s informativeness.

-The term “behavior towards HPV spread” is vague; it could be clarified as “sexual behaviors contributing to HPV transmission.”

Conclusion

-Highlight implications more concretely in the conclusion, e.g., specific interventions.

-The phrase “There is lack of awareness” is grammatically awkward and should be “There is a lack of awareness” or “Awareness is limited.”

-The conclusion is brief and does not emphasize implications for practice or policy, which is critical for a public health study.

-The term “males” is inconsistent with “male partners” used elsewhere; “male partners” should be used for consistency.

Keywords

-Consider reviewing the keywords to ensure they are all MeSH terms

Background

-Streamline the background to ensure that each paragraph is comprehensive and communicates a single idea. For example, paragraph one should only report about the magnitude and burden of cervical cancer from global perspectives to local perspectives, then paragraph two should report what has been done about the problem.

-Contextualize Kenya’s Cervical cancer burden by adding recent national prevalence data

- Streamline the literature review to reduce redundancy, remember to put appropriate intext citation and avoid including outdated references.

-Provide a clear research question or objectives at the end.

-It seems like the references do not match what has been reported in the text.

Methodology

-The methodology section is poorly arranged, please revise it entirely and reorganize

-Revise the study design description to clearly articulate the approach and design used with appropriate citation on why you employed such approach and design as far as context of your study is concerned.

-The study setting is not described comprehensively, rather it is stated out. Please revise and explain why you selected that setting and not others

-The sample and sampling procedure are not well articulated. Please describe them in detail, so that the readers can understand well the protocols undertaken for your study.

-Your tables are not cross-referenced. Consider cross-referencing your tables after you have re-analyzed your data

Results

-In qualitative study, we prefer to use the term “findings” rather than “results”

-The analysis is poorly performed and there is an overlap between the themes and redundancy of information including quotations. Please reanalyze again your transcript according to thematic analysis as described by Braun and Clarke, https://www.tandfonline.com/doi/abs/10.1191/1478088706QP063OA?casa_token=0BCmSV7WAq0AAAAA:Q0509jaFWXmyxHe6nR9BqmaP6Yr4-0W757nPcFpNqWgLGNGVS0nK592lpNS_922_s47pP8TT3XSbbA

-After you have reanalyzed your data, and read to understand how qualitative findings are described and reported.

Discussion

-The discussion is more centered on repeating more of the results of the study, which is not acceptable.

-After you have re-analyzed your data, rewrite the discussion section to entail the following aspects;

Summarize your findings in relation with your study objectives

Compare and contrast your findings with other findings from diverse researches

Justify the congruency and incongruency for the comparisons done per each key and significant finding

Provide an implication (clinical, policy etc.,) for each key and significant finding

-Revise the study limitations to include methodological-related constraints

-Revise the recommendations to reflect what you found please

-The conclusions provided is irrelevant to your findings. Revise the conclusions to stem from your findings and remember that conclusion is not a summary of your study rather an analysis of your study with greater focus on what you found.

References

-The references are poorly reported. Why the authors names capitalized?

-Does the citation match the journal format?

-The references seem to be unrelated to the what has been reported and described in the whole manuscript.

-Consider the use of reference managers to cite appropriately and maintain scientific integrity of your work.

**Reviewer #2: ** The manuscript covers a very imortant area. If authors work on the comments, the study will be very insightful. All the comments provided can be worked to improve the manuscript. I have attached a separate file with comments

**Do you want your identity to be public for this peer review?** For information about this choice, including consent withdrawal, please see our Privacy Policy

Reviewer #1: No

Reviewer #2: No

---

## [Author Response · Author response to Decision Letter 1]

22 Jul 2025

We appreciate the editor and reviewers’ valuable feedback, and we are pleased to incorporate these recommendations. We have resubmitted the revised manuscript that addresses all concerns point-by-point.

---

## [Editor Report · Decision Letter 1]

28 Jul 2025

Dear Dr. MWANGI,

Thank you for submitting your manuscript to PLOS ONE. After careful consideration, we feel that it has merit but does not fully meet PLOS ONE’s publication criteria as it currently stands. Therefore, we invite you to submit a revised version of the manuscript that addresses the points raised during the review process.

We look forward to receiving your revised manuscript.

Kind regards,

Morufu Olalekan Raimi, Ph.D

Academic Editor

PLOS ONE

Journal Requirements:

**Additional Editor Comments:**

The decision is to Accept with Minor Revisions.

The revised manuscript demonstrates substantial improvement and effectively addresses the key concerns raised by the editor. The title and abstract revisions enhance clarity and consistency, particularly in the correct use of terms such as “male partners” and clear articulation of their roles and barriers. Consistent terminology throughout strengthens the scholarly tone, especially concerning transmission, prevention, and control of cervical cancer. The abstract now succinctly highlights major themes, providing readers with a precise overview of the study focus and findings.

In the background and methods sections, the manuscript’s logical flow and rigor have been notably enhanced. The restructuring from a broad global context to a focused regional Kenyan perspective clarifies the study’s relevance and scope, while eliminating redundancy increases readability. The methodology is clearly detailed, with precise descriptions of qualitative design, participant sampling, and data collection. Clear reporting of participant numbers, study settings, and rationale contribute to methodological transparency. Explanation of data saturation for each participant group adds to the robustness of the study design.

The findings section aligns well with qualitative research conventions, with themes and subthemes informed by Braun and Clarke’s analytic framework. Reduced thematic overlap and clear linkage between themes and study objectives improve coherence and reader comprehension. Inclusion of participant quotes substantiates the analysis and enriches the narrative. In the discussion, the authors thoughtfully relate findings to existing literature and practical implications, and responsibly acknowledge methodological limitations, including constraints of qualitative generalizability, framing the study’s contribution appropriately.

Finally, language polishing and formatting refinements have significantly improved the manuscript’s professional presentation. Though minor phrasing and grammatical adjustments remain, these do not detract from the overall quality and clarity. Synthesizing the editorial and reviewer feedback, my final recommendation for acceptance with minor revisions is well-justified. The manuscript is now well-positioned to make a valuable contribution to the literature on male involvement in cervical cancer prevention in Kenya, pending completion of the suggested editorial refinements for optimal readability and style adherence.

The manuscript has substantially addressed all major revisions requested by the editor and is now suitable for publication in PLOS ONE, pending final proofreading for minor grammatical refinements and editorial polishing as outlined in the detailed assessment.

Kind Regards

Prof. Morufu Olalek

---

## [Author Response · Author response to Decision Letter 2]

11 Aug 2025

We have now reviewed the document to verify and adjust references and citations as advised. We have also made changes to grammatical errors in the entire document

---

## [Editor Report · Decision Letter 2]

29 Aug 2025

Exploring the Roles of Male partners in the Transmission, Prevention and control of Cervical Cancer in Central Kenya: A Qualitative Study

PONE-D-25-21773R2

Dear Author,

We’re pleased to inform you that your manuscript has been judged scientifically suitable for publication and will be formally accepted for publication once it meets all outstanding technical requirements.

Kind regards,

Morufu Olalekan Raimi, Ph.D

Academic Editor

PLOS ONE

Additional Editor Comments (optional):

PONE-D-25-21773

"Exploring the Roles of Male partners in the Transmission, Prevention and control of Cervical Cancer in Central Kenya: A Qualitative Study"

The manuscript has been significantly improved beginning with the title and abstract, which have been revised for clarity and consistency. The title now uniformly uses “male partners” and concludes with “A Qualitative Study,” while the abstract has been restructured to correct grammatical errors, align terminology such as “transmission, prevention, control,” and clarify the roles of male partners with a stronger focus on barriers and challenges. These changes ensure the title and abstract meet the journal’s standards.

In addition, the background section now offers a logical and concise narrative, beginning with the global and regional burden of cervical cancer and narrowing focus to Kenya and other low- and middle-income countries (LMICs). Redundancies have been reduced, and the research question and objectives are clearly stated at the conclusion of this section, thus providing a clear rationale and context for the study. The methods section has also been thoroughly refined with enhanced transparency: the qualitative research design is clearly described with details on semi-structured interviews and focus groups, sampling approaches are well differentiated, participant numbers reconciled, and the study setting in Central Kenya justified. Data saturation is explicitly addressed, and acronyms are properly defined on first use, making the methodology replicable.

Moreover, the findings section has been renamed and substantially strengthened by reanalyzing data using Braun and Clarke’s thematic analysis framework. Themes and subthemes are logically developed and each is supported by participant quotes, with distinctions drawn clearly between perceived roles and challenges, reducing redundancy. The discussion cohesively summarizes key findings in relation to objectives, situates the study within existing literature on Sub-Saharan Africa and LMICs, and justifies observed similarities and differences. Importantly, policy and clinical implications—such as involving men in cervical cancer screening awareness campaigns—are emphasized, while methodological limitations, including generalizability concerns and potential social desirability bias, are openly acknowledged.

Finally, the conclusion directly reflects the study findings, highlighting implications for practice and policy, specifically the need for gender-inclusive interventions promoting male partner involvement in screening programs. The manuscript’s references have been updated and formatted to adhere to PLOS ONE style, incorporating recent literature. Overall language and formatting issues have been addressed through proofreading, resulting in improved readability and consistent terminology throughout. The authors have successfully responded to all editorial comments, resolving previous concerns related to methodology and structure. Hence, the manuscript now meets PLOS ONE’s standards for scientific validity, transparency, and relevance, warranting acceptance.

Recommendation: Accepted

Sincerely,

Dr. Morufu Olalekan RAIMI,

BSc, (Geography. and Environmental Management), Diploma. (Environmental Health), M.Sc. Environmental Health Management), M.Phil. (Environmental Health Science), P.hD (Environmental Health Science), MNES, REHO, LEHO, FAIWMES

Environmental Health Consultant/Lecturer at Federal University Otuoke, Bayelsa State. Nigeria.

Environmental Health Consultant to United Nations Economic Commission for Europe (UNECE) Expert Group on Resources Management (EGRM). Geneva, Switzerland.

Research Consultant to Bayelsa State Primary Health Care Board.

Former Technical Adviser to the Executive Secretary, Bayelsa State Primary Health Care Board.

Former Director, Advocacy, Communication and Social Mobilization, Bayelsa State Primary Health Care Board.

Program Manager, Centre for Niger Delta Studies and Sustainability (CNDSS), Federal University Otuoke, Bayelsa State.

Deputy Director, Niger Delta Institute for Emerging and Re-emerging Infectious Diseases (NDIERID), Federal University Otuoke, Bayelsa State.

Reviewer to National Science Foundation (NSF) Graduate Research Fellowship Program (GRFP)

https://publons.com/a/1479339/

https://ssrn.com/author=2891311

ORCID iD: https://orcid.org/0000-0001-5042-6729

Web of Science Researcher ID: https://publons.com/a/1479339/

Website: https://ssrn.com/author=2891311;
https://www.growkudos.com/profile/morufu_raimi;
https://sciprofiles.com/profile/Morufuolalekanraimi;
https://livedna.org/234.27529

https://scholar.google.com/citations?user=nRBW82AAAAAJ&hl=en.

https://theconversation.com/profiles/morufu-olalekan-raimi-1520774
---

## [Editor Report · Acceptance letter]

PONE-D-25-21773R2

PLOS ONE

Dear Dr. Mwangi,

I'm pleased to inform you that your manuscript has been deemed suitable for publication in PLOS ONE. Congratulations! Your manuscript is now being handed over to our production team.

Kind regards,

on behalf of

Prof Morufu Olalekan Raimi

Academic Editor

PLOS ONE